# Optimization of Different Factors for Initiation of Somatic Embryogenesis in Suspension Cultures in Sandalwood (*Santalum album* L.)

**Manoj Kumar Tripathi** [1,2,*], **Niraj Tripathi** [3], **Sushma Tiwari** [2], **Gyanendra Tiwari** [1,4], **Nishi Mishra** [2,5], **Dilip Bele** [1], **Rajesh Prasad Patel** [6], **Swapnil Sapre** [5] and **Sharad Tiwari** [5]

1 Horticultural Biotechnology Laboratory, College of Horticulture, Mandsaur-458001, RVS Agricultural University, Gwalior 474002, India; drgyanendratiwari@gmail.com (G.T.); dilipbele2021@gmail.com (D.B.)
2 Department of Plant Molecular Biology & Biotechnology, College of Agriculture, RVS Agricultural University, Gwalior 474002, India; sushma2540@gmail.com (S.T.); nishimishra554@gmail.com (N.M.)
3 Directorate of Research Services, Jawaharlal Nehru Agricultural University, Jabalpur 482004, India; tripathi.niraj@gmail.com
4 Department of Plant Physiology, Jawaharlal Nehru Agricultural University, Jabalpur 482004, India
5 Biotechnology Centre, JN Agricultural University, Jabalpur 482004, India; Swapnil.spr@gmail.com (S.S.); shtiwari@gmail.com (S.T.)
6 Department of Plant Pathology, College of Horticulture, Mandsaur–458001, RVS Agricultural University, Gwalior 474002, India; rajeshpatel179@gmail.com
* Correspondence: drmanojtripathi64@gmail.com

**Abstract:** *Santalum album* (L.) is a prized tropical tree species of high therapeutic and industrial importance. The wood of these naturally grown plants is extensively harvested to acquire therapeutically important metabolite santalol and be used for additional functions such as in wood statuette industries. Due to high demand, it is crucial to maintain a sufficient plant population. An easy protocol for establishing cell suspension culture initiated from the loose embryogenic callus mass of sandalwood was realized by shifting 6–8-week-old morphogenic calli acquired from the mature embryonic axis and cotyledon explant cultures in fluid media. The asynchronous embryogenic cultures were sloughed with clumps of flourishing cell clumps and embryos of various progressive phases along with diffident non-embryogenic tissues. The frequency of embryo proliferation was evidenced to determinethe expansion pace of embryogenic masses under diverse conditions. The intonation of initiation and creation of cell suspension was under the directive of the influence of exogenous plant growth regulators amended in the nutrient medium at different concentrations and combinations. Maximum relative growth rate (386%) and clumps/embryoids in elevated integers (321.44) were accomplished on MS nutrient medium fortified with 2.0 mg L$^{-1}$ 2,4-D in association with 0.5 mg L$^{-1}$ BA and 30.0 g L$^{-1}$ sucrose raised from mature embryonic axis-derived calli. Plantlet regeneration in higher frequency (84.43%) was evidenced on MS medium amended with 1.0 mg L$^{-1}$ each of TDZ and GA$_3$ in conjunction with 0.5 mg L$^{-1}$ NAA and 20.0 g L$^{-1}$ sucrose. Mature embryonic axis-derived calli were found to be constantly better than mature cotyledon-derived calli for raising profitable and reproducible cell suspension cultures. Regenerants displayed normal growth and morphology and were founded successfully in the external environment after hardening.

**Keywords:** cell clumps; cell suspension culture; plantlet regeneration; *Santalum album*; somatic embryogenesis

## 1. Introduction

*Santalum album* (L.) is a prized tropical tree species belonging to Santalaceae [1]. It is an occupant of the Indian subcontinent. synonymous with prehistoric Indian culture and convention [2]. Santalol, a secondary metabolite in *S. album*, is identified as sandalwood oil, being exceedingly priced in the perfumery traffic owing to its sugary, constant perfume and

adhesive properties [3]. It also finds widespread submissions in conventional therapeutic schemes, for instance, Ayurveda, and is achieving escalating significance in contemporary pharmacological research as a source of anticancer [4], anti-*Helicobacter pylori* (Marsh [5] and antiviral [6] ingredients. The wood of naturally grown plants is extensively harvested to acquire santalol and for other reasons, such as to supply wood statuette industries and native medicine. Even though the species promulgated by seeds in nature, the pace of success is only 20% [7,8]. The viability of seeds is gone within 6–9 months. Traditional breeding of sandalwood for incorporating new genetic information may be costly and complicated due to longgeneration, sexual incongruity and heterozygous character [9]. These factors are responsible for the obliteration of *S. album* trees in India.

In vitro culture practices may be applied to overcome complications of conventional propagation techniques by microcloning of better lines. Besides, the destruction caused by the pandemic spike disease, hemiparasitic nature and deliberate-cultivating character of the plant dictated investigation in the direction of the extension of modern unconventional tools of in vitro propagation, namely callusing, in the early 1960s [10]. Later on, attempts of somatic embryo invention [11] and maturation in bioreactors [12,13] were successfully documented. In addition, the uniform and synchronous suspension cultures make them suitable for the creation of phytochemicals at a large scale. Substantial work has been done in sandalwood by several scientists throughout the world so far by employing an array of explants for inducing morphogenesis followed by plantlet regeneration with varying level of accomplishment; for instance, embryos [10,14], hypocotyls [14–16], shoot tips [17,18], nodal segments [19–22], leaf discs [23,24], seedlings [25], endosperms [15,16], cotyledons [14] and protoplasts [26] have been used jointly with cell suspension cultures [11–13,27].

Cell suspension culture may be used to select valuable mutant cell lines through in vitro selection [28–31] for useful secondary metabolites [32–34]. It could also be employed for genetic transformation. In cell suspension culture, the selection system is meticulous, and expansion is faster compared to tissue inoculated on semi-solid media because of the elevated medium-to-tissue contact. Usually, morphogenetic tissues at an early ontogenetic phase are utilized to commence embryogenic cell suspensions. In the present investigation, an attempt has been made to discover the best performing explants and PGRs and determine their most advantageous concentrations along with other culture conditions displaying in vitro embryogenesis in a high ratio through cell suspension cultures of an elite cultivar of sandalwood now under cultivation in Madhya Pradesh and adjacent regions to avoid additional setback of adaption.

## 2. Materials and Methods

### 2.1. Experimental Materials

A local cultivar of Indian sandalwood *Santalum album* (L.) growing naturally in the Malwa region of Madhya Pradesh, India, was selected for the raising callus cultures from the mature embryonic axis and cotyledon explants and consequently the establishment of embryogenic cell suspension cultures. Seeds were collected from 5–10-year-old plants, planted at Bahadri Farm, College of Horticulture, Mandsaur, Madhya Pradesh, India.

### 2.2. Culture Media

To instigate preliminary research, two different basal media, namely MS [35] and WP [36], were tested to assess enhanced in vitro comeback. In the itinerary of the beginning study, MS nutrient medium was proven more amenable than WP. Consequently, MS was used as a basal medium for final work. Aside from MS basal micro and macro salts, vitamins and agar powder, two diverse auxins (alone), namely 2,4-D and NAA, and three different cytokinins (alone),i.e., BAP, Kn and TDZ, in diverging concentrations in addition to 2,4-D and NAA in amalgamation with BAP and Kn, 30.0 g L$^{-1}$ sucrose and 7.5 g L$^{-1}$ agar were supplemented to formulate MS media for raising callus cultures from the mature embryonic axis and cotyledon explants followed by establishing embryogenic cell suspension cultures. One liter culture media was prepared by adding readymade powder and plant

growth regulators from readymade stock solutions in liquid form, and pH was adjusted to $5.8 \pm 0.1$ with 1N NaOH or HCl. After normalizing the pH, 7.5 g $L^{-1}$ agar was included in the media for semi-solidification. For cell suspension culture experiments, agar was not added to the medium. Liquid warm culture media were poured into culture bottles (50–60 mL/bottle) and culture tubes (15–20 mL/tube) and kept for autoclaving at 121 °C under 15 psi pressure for 20–25 min. Culture media combinations and other ingredients were finalized based on the work accomplished by other scientists and earlier experiences of this laboratory. Readymade MS basal medium, plant growth regulators and supplementary ingredients were acquired from Hi-Media Laboratories, Mumbai, India.

### 2.3. Establishment of Callus Cultures

Mature embryonic axis and cotyledons were excised from mature seed. For this intention, seeds were rinsed with 2% Tween 20 (*v*/*v*) for 15–20 min followed by thoroughly rinsing with flooding tap water for 30 min to eradicate dust and dregs. Then, they were treated with 70% (*v*/*v*) ethyl alcohol for 2 min followed by dipping in 0.5% of Bavistin in permutation with 0.1% $HgCl_2$ for 20 min. Finally, the seeds were washed 4–5 times with sterilized double-distilled water and dipped overnight in disinfected water. Mature embryonic axis was excised from these seeds; however, mature cotyledonexplants were obtained from 14–21-day-old germinated seeds from inoculated seeds in culture tubes supplemented with MS medium without plant growth regulators and 7.5 g $L^{-1}$ agar under dispersed light of 16 mol $m^{-2}$ $s^{-1}$ supplied with white fluorescent lamps. Seven to eight sections of each of the explants were inoculated and closed with Parafilm before incubation under complete darkness at $25 \pm 2$ °C for a week. Afterward, in vitro inoculated explants were transferred to a photoperiod of 8 h dark and 16 h 2000-lux luminance generated by the PAR lamps.

### 2.4. Establishment of Cell Suspension Culture and Somatic Embryogenesis

For establishing friable cell suspension cultures, 6–8-week-old friable calli (1.0g fresh mass) obtained from the mature embryonic axis and cotyledon explant cultures were added to 250 mL Erlenmeyer flasks having 50 mL of MS liquid medium. Callus pieces were strained through a stainless steel mesh (1mm) and were agitated on a horizontal shaker (120 rpm) at $25 \pm 2$ °C in the dark. After 15 days, the cultures were sieved aseptically to remove large clumps, and 10 mL filtrate was added to 40 mL of the fresh medium by restoring the old suspension for subculturing. The relative growth rate was calculated by augmentation in fresh weight following inoculating morphogenic friable calli on diverse liquid medium after 4–6 weeks of initial culture. Cell cultures were examined microscopically within 15 to 35 days for induction of somatic embryoids/cell clumps and determination of developmental pathways.

### 2.5. Plantlet Regeneration

Cell clumps/embryoids of 2.0 to 6.0 mm sizes obtained from 4–6-week-old liquid suspension cultures were put onto semisolid MS medium fortified with diverse concentrations and combinations of PGRs (NAA, BAP, Kn, TDZ and $GA_3$ alone or in combinations), 20.0 g $L^{-1}$ sucrose and 7.5 g $L^{-1}$ agar. Culture tubes were then transferred under $25 \pm 2$ °C temperature and photoperiod regimes of 60 µ mol $m^{-2}$ $s^{-1}$ luminance supplied by cool fluorescent tubes for 16 h. Plantlet regeneration percentages were calculated as numbers of cell clumps/embryoids with plantlets from the sum of cell clumps/embryoids transferred on regeneration medium.

### 2.6. In Vitro Rooting of Regenerants

Regenerants were subsequently transferred to a rooting medium consisting of diverse levels of IBA, NAA, BA, Kn and $GA_3$ alone and in combinations, 15.0 g $L^{-1}$ sucrose and 7.5 g $L^{-1}$ agar for root induction. For rooting, a decreased amount of sucrose was employed

based on experiments accomplished by worldwide researchers and prior experiences of this laboratory.

### 2.7. Acclimatization of Regenerants

Complete plantlets were systematically cleaned under flooding water and were transferred in 2.5 cm root trainers covered with 1:1:1 sand, soil and farmyard manure sterilized arrangement. Root trainers with regenerants were then accommodated in the environmental growth cabinet for 15–20 days under 30 ± 2 °C and 65 ± 5% RH followed by transferring to greenhouse conditions for 25–30 days and polyhouse for 20–25 days and then transplanted in the field.

### 2.8. Experimental Design and Data Analysis

Completely randomized design (CRD) was applied to determine the importance of different culture media arrangements. Each treatment had two replications. For each replication, around 400–600 cell clumps/embryoids were plated on each media. Arc-sine and/or log transformation was adapted as per prerequisite before analysis of data. The data were analyzed as per Snedecor and Cochran [37].

## 3. Results

### 3.1. Establishment of Callus Cultures

Callus culture was established from the mature embryonic axis and cotyledon explants as per protocol standardized by Tripathi et [14]. The initial behavior of inoculated embryonic axis and cotyledons was parallel within 5–7 days and generally sovereign from explant and culture media. Every explant turned out to be enlarged, and callus induction was not evident in the first few days. After 7–10 days of inoculation, callus proliferation was apparent from the embryonic axis of plated mature embryos. However, from cultured mature cotyledons, callus initiation took comparatively more time and generally proliferated from the dissected ends following 10–14 days of culture. The calli were light white to moderate yellow-greenish in color, small to medium in size and friable to compact in texture. Friable calli initiated morphogenic calli in lesser numbers conversely; such calli were more amenable for establishing cell suspension culture. In contrast, compact nodular callus demonstrated higher morphogenetic competence but was not suitable for initiating cell suspension cultures. Induction media MS2D (MS + 2.0 mg $L^{-1}$ 2,4-D) proved adequate for callus initiation and proliferation.

### 3.2. Cell Suspension Culture Establishment and Somatic Embryo Induction

To initiate cell suspension cultures, ~1 g friable morphogenic calli was shifted in a liquid medium. When agitated in liquid culture, friable calli were effortlessly broken and separated into clumps of ~2–5 mm sizes. Later, shaking shattered these clumps into small cell aggregates (Figure 1A,B). Calli acquired from the mature embryonic axis (Figure 1A) responded better as compared to calli derived from mature cotyledons (Figure 1B) for growth rate and clump development leading to in vitro embryogenesis (Tables 1–3). When examined under a stereomicroscope, cells obtained from the suspension cultures were undifferentiated meristem-like by way of opaque cytoplasm and prominent nuclei. Suspension cultures obtained from morphogenic callus fabricated embryoids of different sizes and shapes that, differently from the typical developmental pathway, began from globular stage (Figure 1C) and proceeded to heart stage (Figure 1D) followed by torpedo stage (Figure 1E). Sometimes, bipolar embryoids (Figure 1F) that enlarged and germinated after maturation also formed (Figure 1G). A different pathway adopted by the cell suspension culture created multiple embryoids through the course of secondary embryogenesis on the surface of primary embryoids (Figure 1H). The clumps of increasing globular embryos were typically white to yellowish and of varying sizes. Individual clumps were multilobed, with each lobe symbolizing a premature-phase globular embryo. Somatic embryos were connected at their bottoms to cultures, and phenotypic manifestation indicated that they arose from the facade

of primary embryos progressively converting into shoots (Figure 1I). A few embryoids twisted into multipolar configuration (Figure 1J), displaying manifold gammo-rhizogenesis from a distinct embryoid (Figure 1K). A 4–6-week culture phase was obligatory to decide which route would be followed by cell suspension cultures. Plantlet formation was obtained on regeneration medium after transferring cell clumps/embryoids from 6–8-week-old suspension cultures (Figure 1L). Shootlets were subsequently re-cultured on the rooting medium (Figure 1M). Regenerants of 2–3 cm in length were transferred to a greenhouse (Figure 1N), followed by the net house for 25–30 days (Figure 1O) for hardening, and ultimately to the field (Figure 1P). Surviving regenerants were assessed phenotypically on the origin of their facade. Even if the traits were not counted numerically, regenerated shootlets obtained were usually similar to their mother plants.

**Table 1.** Influence of diverse auxins (alone) in varying levels on growth of cell suspension cultures.

| Auxin ▼ | Conc. Mg L$^{-1}$ | Embryogenic Suspension Cultures | | | | | |
|---|---|---|---|---|---|---|---|
| | | Mature Embryonic Axis-Derived Calli | | | Mature Cotyledon-Derived Calli | | |
| | | Increase in FW (g) * | RG (%) | Number(s) of Embryoids/Flask ** | Increase in FW (g) * | RG (%) | Number(s) of Embryoids/Flask ** |
| Control | 0.0 | * 1.00 ± 0.08 | 100 | | 1.00 ± 0.09 | 100 | |
| MS.1D | 0.1 | 1.14 ± 0.10 | 114 | 137.24 [l] (2.14) | 1.12 ± 0.08 | 112 | 123.44 [m] (2.10) |
| MS.5D | 0.5 | 1.44 ± 0.20 | 144 | 152.48 [j] (2.19) | 1.35 ± 0.16 | 135 | 148.63 [j] (2.17) |
| MSD | 1.0 | 1.56 ± 0.24 | 156 | 168.20 [h] (2.23) | 1.49 ± 0.20 | 149 | 163.44 [h] (2.22) |
| MS2D | 2.0 | 2.40 ± 0.30 | 240 | 248.64 [d] (2.40) | 2.24 ± 0.26 | 224 | 242.63 [d] (2.39) |
| MS3D | 3.0 | 2.72 ± 0.36 | 272 | 282.42 [a] (2.45) | 2.66 ± 0.30 | 266 | 278.44 [a] (2.45) |
| MS4D | 4.0 | 2.98 ± 0.38 | 298 | 275.82 [b] (2.44) | 2.87 ± 0.32 | 287 | 262.49 [b] (2.42) |
| MS5D | 5.0 | 1.48 ± 0.14 | 148 | 124.62 [n] (2.10) | 1.37 ± 0.12 | 137 | 120.75 [n] (2.09) |
| MS.1N | 0.1 | 1.16 ± 0.12 | 116 | 132.14 [m] (2.12) | 1.12 ± 0.12 | 112 | 127.14 [l] (2.11) |
| MS.5N | 0.5 | 1.44 ± 0.18 | 144 | 142.44 [k] (2.16) | 1.28 ± 0.16 | 128 | 138.62 [k] (2.14) |
| MSN | 1.0 | 1.68 ± 0.24 | 168 | 160.49 [i] (2.21) | 1.49 ± 0.22 | 149 | 154.15 [i] (2.19) |
| MS2N | 2.0 | 2.16 ± 0.28 | 216 | 228.66 [f] (2.36) | 2.10 ± 0.26 | 210 | 224.18 [f] (2.35) |
| MS3N | 3.0 | 2.80 ± 0.32 | 280 | 252.48 [c] (2.40) | 2.40 ± 0.30 | 240 | 248.44 [c] (2.40) |
| MS4N | 4.0 | 2.92 ± 0.36 | 292 | 245.26 [e] (2.39) | 2.88 ± 0.34 | 288 | 240.61 [e] (2.38) |
| MS5N | 5.0 | 1.66 ± 0.20 | 166 | 188.36 [g] (2.28) | 1.47 ± 0.18 | 147 | 182.73 [g] (2.26) |
| Mean | | | | 195.66 (2.27) | | | 189.69 (2.26) |
| CD$_{0.05}$ | | | | 0.0140 | | | 0.0107 |

Evaluation was made after 28 days in culture. Initial fresh weight was taken 1.0 g per flask containing 50 mL liquid medium. **FW**: fresh weight; **RG**: relative growth. * Mean of five readings ± standard deviation. ** Figures in parentheses are transformed values (log transformation). Values within a column followed by different letters significantly differ at 5% probability level.

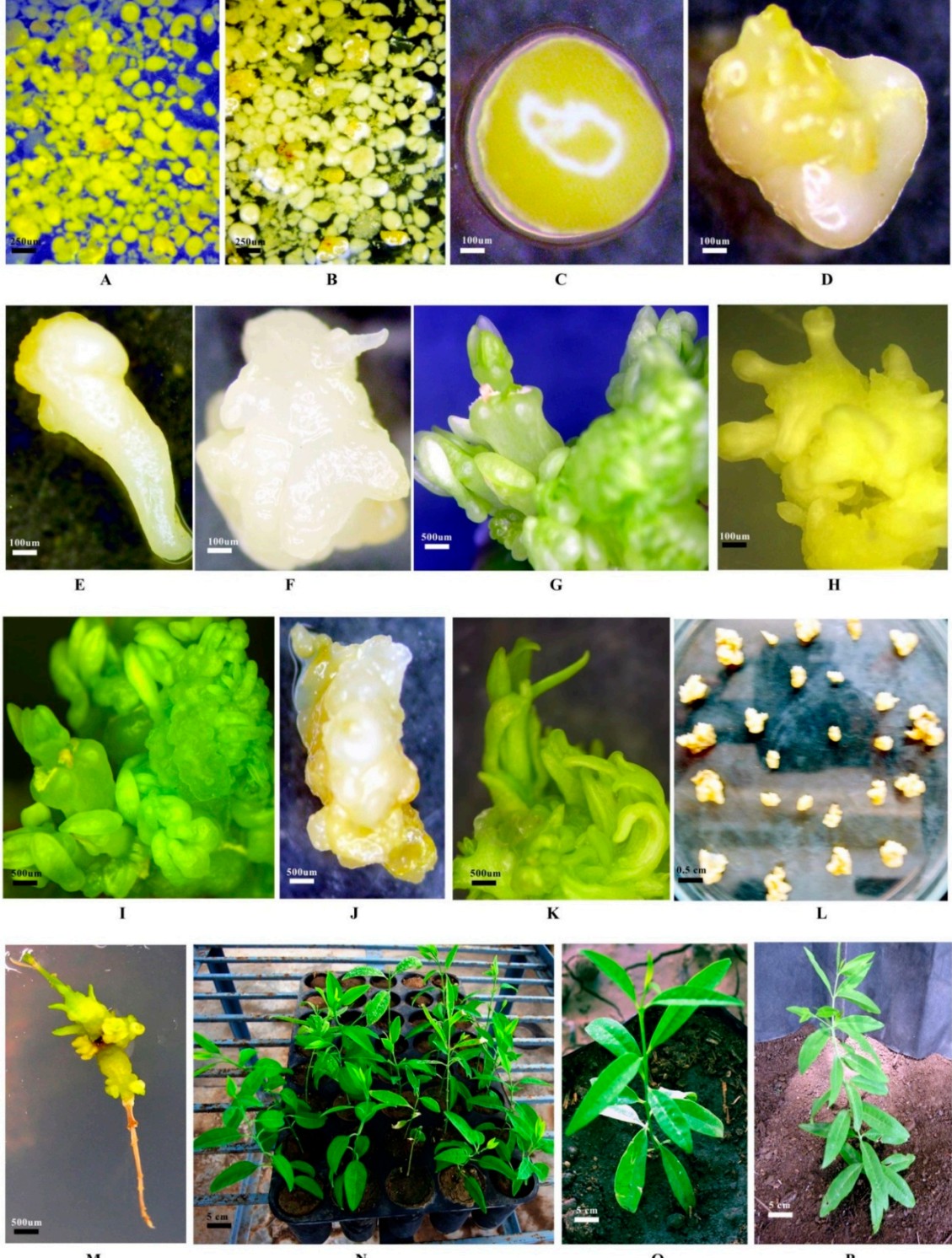

**Figure 1.** Plant regeneration from cell suspension culture in *Santalum album* L.: (**A**) initiation of cell clumps and embryoid formation raised from embryogenic calli of mature embryonic axis; (**B**) initiation of cell clumps and embryoid formation derived from embryogenic calli of mature cotyledons; (**C**) globular stage embryoid; (**D**) heart stage embryoid; (**E**) torpedo stage embryoid; (**F**) typical bipolar embryoid; (**G**) germination of bipolar embryoid; (**H**) multiple embryoids; (**I**) germinationof multiple embryoids; (**J**) typical multipolar embryoid; (**K**) germination of multipolar embryoid; (**L**) embryoid/cell clumps transferred on regeneration medium; (**M**) rooting of regenerants; (**N**) regenerants transferred in the greenhouse for hardening; (**O**) regenerants subjected to the net house for hardening; (**P**) regenerants shifted in the field for hardening.

**Table 2.** Influence of diverse cytokinins (alone) in varying levels on growth of embryogenic cell suspension cultures.

| Cytokinin ▼ | Conc. Mg L$^{-1}$ | Embryogenic Suspension Cultures | | | | | |
|---|---|---|---|---|---|---|---|
| | | Mature Embryonic Axis-Derived Calli | | | Mature Cotyledon-Derived Calli | | |
| | | Increase in FW (g) * | RG (%) | Number(s) of Embryoids/Flask ** | Increase in FW (g) * | RG (%) | Number(s) of Embryoids/Flask ** |
| Control | 0.0 | 1.00 ± 0.06 | 100 | | 1.00 ± 0.04 | 100 | |
| MS.1B | 0.1 | 1.14 ± 0.12 | 114 | 129.22 [p] (2.11) | 1.16 ± 0.11 | 116 | 128.64 [n] (2.11) |
| MS.5B | 0.5 | 1.28 ± 0.16 | 128 | 157.62 [i] (2.20) | 1.23 ± 0.13 | 123 | 150.48 [i] (2.18) |
| MSB | 1.0 | 1.34 ± 0.18 | 134 | 172.43 [f] (2.24) | 1.30 ± 0.16 | 130 | 166.18 [f] (2.23) |
| MS2B | 2.0 | 1.42 ± 0.20 | 142 | 185.15 [e] (2.27) | 1.36 ± 0.18 | 136 | 180.17 [e] (2.26) |
| MS3B | 3.0 | 1.22 ± 0.14 | 122 | 163.81 [h] (2.22) | 1.16 ± 0.12 | 116 | 158.24 [h] (2.20) |
| MS4B | 4.0 | CC | - | 135.55 [m] (2.14) | CC | - | 130.24 [n] (2.12) |
| MS5B | 5.0 | CC | - | 123.61 [s] (2.10) | CC | - | 119.16 [s] (2.08) |
| MS.1Kn | 0.1 | 1.34 ± 0.14 | 134 | 187.44 [d] (2.28) | 1.27 ± 0.12 | 127 | 180.22 [d] (2.26) |
| MS.5Kn | 0.5 | 1.41 ± 0.18 | 141 | 192.26 [c] (2.29) | 1.45 ± 0.16 | 145 | 187.69 [c] (2.28) |
| MSKn | 1.0 | 1.50 ± 0.20 | 150 | 197.82 [b] (2.30) | 1.43 ± 0.18 | 143 | 191.24 [b] (2.29) |
| MS2Kn | 2.0 | 1.78 ± 0.24 | 178 | 215.38 [a] (2.34) | 1.69 ± 0.22 | 169 | 211.43 [a] (2.33) |
| MS3Kn | 3.0 | 1.62 ± 0.19 | 162 | 153.16 [j] (2.19) | 1.54 ± 0.18 | 154 | 148.28 [j] (2.17) |
| MS4Kn | 4.0 | 1.44 ± 0.17 | 144 | 133.46 [n] (2.13) | 1.31 ± 0.15 | 131 | 128.44 [o] (2.11) |
| MS5Kn | 5.0 | HR | - | 124.66 [q] (2.10) | HR | - | 119.43 [r] (2.08) |
| MS.1Td | 0.1 | 1.34 ± 0.14 | 134 | 167.28 [g] (2.23) | 1.27 ± 0.12 | 127 | 159.33 [g] (2.21) |
| MS.2Td | 0.2 | 1.43 ± 0.16 | 143 | 142.48 [k] (2.16) | 1.37 ± 0.14 | 137 | 136.20 [k] (2.14) |
| MS.3Td | 0.3 | 1.25 ± 0.12 | 125 | 138.49 [l] (2.14) | 1.21 ± 0.11 | 121 | 132.22 [l] (2.12) |
| MS.4Td | 0.4 | 1.22 ± 0.10 | 122 | 130.14 [o] (2.12) | 1.16 ± 0.10 | 116 | 126.34 [p] (2.10) |
| MS.5Td | 0.5 | 1.21 ± 0.09 | 121 | 124.22 [r] (2.10) | 1.23 ± 0.11 | 123 | 120.41 [q] (2.08) |
| MSTd | 1.0 | 1.15 ± 0.08 | 115 | 120.23 [t] (2.08) | 1.18 ± 0.09 | 118 | 114.38 [t] (2.06) |
| MS2Td | 2.0 | 1.11 ± 0.08 | 111 | 117.44 [u] (2.07) | 1.12 ± 0.09 | 112 | 113.78 [u] (2.06) |
| MS3Td | 3.0 | CC | - | 112.92 [v] (2.06) | CC | - | 108.26 [v] (2.04) |
| MS4Td | 4.0 | CC | - | 108.44 [w] (2.04) | CC | - | 106.26 [w] (2.03) |
| MS5Td | 5.0 | CC | - | 106.66 [x] (2.03) | CC | - | 104.26 [x] (2.02) |
| Mean | | | | 147.49 (2.16) | | | 142.55 (2.14) |
| CD$_{0.05}$ | | | | 0.0129 | | | 0.0190 |

Evaluation was made after 28 days in culture. Initial fresh weight was taken 1.0 g per flask containing 50 mL liquid medium. **FW**: fresh weight; **RG**: relative growth. * Mean of five readings ± standard deviation. ** Figures in parentheses are transformed values (log transformation). Values within a column followed by different letters significantly differ at 5% probability level. **HR**: hairy root; **CC**: compact callus.

**Table 3.** Cumulative influence of supplemented cytokinins and auxins in varying levels and combinations on growth of sandalwood cell suspension cultures.

| Culture Medium ▼ | Growth Regulators mg L$^{-1}$ | | | | Embryogenic Suspension Cultures | | | | | |
|---|---|---|---|---|---|---|---|---|---|---|
| | | | | | Mature Embryonic Axis-Derived Calli | | | Mature Cotyledon-Derived Calli | | |
| | 2,4-D | NAA | BA | Kn | Increase in FW (g) * | RG (%) | Number(s) of Embryoids/Flask ** | Increase in FW (g) * | RG (%) | Number(s) of Embryoids/Flask ** |
| Control | - | - | - | - | 1.00 ± 0.10 | 100 | | 1.00 ± 0.12 | 100 | |
| MS.1D.5B | 0.1 | - | 0.5 | - | 1.36 ± 0.16 | 136 | 162.21 [n] (2.21) | 1.28 ± 0.14 | 128 | 161.23 [m] (2.21) |
| MS.5D.5B | 0.5 | - | 0.5 | - | 1.45 ± 0.18 | 145 | 182.41 [i] (2.26) | 1.36 ± 0.15 | 136 | 170.66 [k] (2.24) |
| MSD.5B | 1.0 | - | 0.5 | - | 1.52 ± 0.20 | 152 | 182.63 [h] (2.27) | 1.46 ± 0.18 | 146 | 179.40 [h] (2.26) |
| MS2D.5B | 2.0 | - | 0.5 | - | 3.86 ± 0.34 | 386 | 321.44 [a] (2.51) | 3.72 ± 0.30 | 372 | 318.62 [a] (2.51) |
| MS3D.5B | 3.0 | - | 0.5 | - | 3.34 ± 0.24 | 334 | 294.42 [b] (2.47) | 3.26 ± 0.22 | 326 | 290.47 [b] (2.47) |
| MS4D.5B | 4.0 | - | 0.5 | - | 202 ± 0.22 | 202 | 234.12 [e] (2.37) | 1.98 ± 0.20 | 198 | 230.34 [e] (2.37) |
| MS.1D.5Kn | 0.1 | - | - | 0.5 | 1.40 ± 0.16 | 140 | 144.13 [s] (2.16) | 1.32 ± 0.14 | 132 | 141.23 [s] (2.15) |
| MS.5D.5Kn | 0.5 | - | - | 0.5 | 1.58 ± 0.20 | 158 | 155.13 [o] (2.20) | 1.50 ± 0.18 | 150 | 150.63 [p] (2.18) |
| MSD.5Kn | 1.0 | - | - | 0.5 | 2.32 ± 0.28 | 232 | 168.40 [l] (2.23) | 2.24 ± 0.26 | 224 | 165.38 [l] (2.22) |
| MS2D.5Kn | 2.0 | - | - | 0.5 | 2.04 ± 0.22 | 204 | 279.48 [c] (2.45) | 2.88 ± 0.21 | 288 | 275.42 [c] (2.44) |
| MS3D.5Kn | 3.0 | - | - | 0.5 | 1.74 ± 0.18 | 174 | 248.52 [d] (2.40) | 1.62 ± 0.17 | 162 | 242.50 [d] (2.39) |
| MS4D.5Kn | 4.0 | - | - | 0.5 | HR | - | 198.36 [f] (2.30) | HR | - | 194.40 [f] (2.29) |
| MS.1N.5B | - | 0.1 | 0.5 | - | 1.24 ± 0.16 | 124 | 125.25 [w] (2.10) | 1.20 ± 0.14 | 120 | 120.17 [w] (2.08) |
| MS.5N.5B | - | 0.5 | 0.5 | - | 1.34 ± 0.18 | 134 | 134.16 [v] (2.13) | 1.32 ± 0.16 | 132 | 130.17 [v] (2.12) |
| MSN.5B | - | 1.0 | 0.5 | - | 1.48 ± 0.18 | 148 | 154.66 [p] (2.19) | 1.36 ± 0.18 | 136 | 152.28 [o] (2.19) |
| MS2N.5B | - | 2.0 | 0.5 | - | 2.58 ± 0.22 | 258 | 178.37 [k] (2.26) | 2.52 ± 0.20 | 252 | 175.29 [i] (2.25) |
| MS3N.5B | - | 3.0 | 0.5 | - | 2.28 ± 0.16 | 228 | 185.40 [g] (2.27) | 2.22 ± 0.14 | 222 | 182.39 [g] (2.27) |

**Table 3.** *Cont.*

| Culture Medium ▼ | Embryogenic Suspension Cultures | | | | | | | | |
|---|---|---|---|---|---|---|---|---|---|
| | Growth Regulators mg L$^{-1}$ | | | | Mature Embryonic Axis-Derived Calli | | | Mature Cotyledon-Derived Calli | | |
| | 2,4-D | NAA | BA | Kn | Increase in FW (g) * | RG (%) | Number(s) of Embryoids/Flask ** | Increase in FW (g) * | RG (%) | Number(s) of Embryoids/Flask ** |
| MS4N.5B | - | 4.0 | 0.5 | | 1.22 ± 0.11 | 122 | 152.74 [q] (2.19) | 1.21 ± 0.10 | 121 | 150.37 [q] (2.18) |
| MS.1N.5Kn | - | 0.1 | - | 0.5 | 1.18 ± 0.10 | 118 | 120.39 [x] (2.08) | 1.14 ± 0.09 | 114 | 115.24 [x] (2.06) |
| MS.5N.5Kn | - | 0.5 | - | 0.5 | 136 ± 0.18 | 136 | 138.42 [t] (2.14) | 1.28 ± 0.16 | 128 | 132.46 [t] (2.13) |
| MSN.5Kn | - | 1.0 | - | 0.5 | 1.50 ± 0.20 | 150 | 162.47 [m] (2.22) | 1.42 ± 0.18 | 142 | 158.38 [n] (2.20) |
| MS2N.5Kn | - | 2.0 | - | 0.5 | 1.20 ± 0.14 | 120 | 180.18 [j] (2.26) | 1.16 ± 0.12 | 116 | 174.22 [j] (2.25) |
| MS3N.5Kn | - | 3.0 | - | 0.5 | HR | - | 145.10 [r] (2.17) | HR | - | 142.26 [r] (2.16) |
| MS4N.5Kn | - | 4.0 | - | 0.5 | CM | - | 135.34 [u] (2.13) | CM | - | 132.33 [u] (2.12) |
| Mean | | | | | | | 182.66 (2.25) | | | 178.58 (2.24) |
| CD$_{0.05}$ | | | | | | | 0.176 | | | 0.0178 |

Evaluation was made after 28 days in culture. Initial fresh weight was taken 1.0 g per flask containing 50 mL liquid medium. **FW**: fresh weight; **RG**: relative growth. * Mean of five readings ± standard deviation. ** Figures in parentheses are transformed values (log transformation). Values within a column followed by different letters significantly differ at 5% probability level. **HR**: hairy root; **CM**: cell morality.

### 3.3. Influence of Plant Growth Regulators on Establishment of Suspension Culture and Somatic Embryo Induction

The variance analysis (Tables 1–4) revealed highly significant ($p < 0.01$) dissimilarities among the manifestation of diverse nutrient media arrangements in idioms of number(s) of embryoids/flask and formation of regenerants. It indicates the existence of a substantial sum of unevenness amongst various culture media amalgamations.

**Table 4.** Combined influence of different plant growth regulators at variable concentrations and combinations on regeneration on plantlets from embryoids/cell clumps raised from embryogenic suspension cultures.

| Culture Media ▼ | Plantlet Regeneration (%) | | | | | | |
|---|---|---|---|---|---|---|---|
| | Plant Growth Regulator mg L$^{-1}$ | | | | | Mature Embryonic Axis -Derived Cell Clumps/Embryoids | Mature Cotyledon-Derived Cell Clumps/Embryoids |
| | BAP | TDZ | Kn | GA$_3$ | NAA | | |
| MS.5B.5G | 0.5 | - | - | 0.5 | - | 52.54 [lm] (46.44) | 46.94 [m] (43.23) |
| MSB.5G | 1.0 | - | - | 0.5 | - | 57.62 [i] (49.36) | 51.82 [k] (46.02) |
| MS2B.5G | 2.0 | - | - | 0.5 | - | 64.28 [gh] (53.28) | 58.48 [i] (49.86) |
| MS.5BG | 0.5 | - | - | 1.0 | - | 58.34 [i] (49.78) | 54.49 [j] (47.56) |
| MSBG | 1.0 | - | - | 1.0 | - | 63.66 [h] (52.91) | 56.14 [j] (48.51) |
| MS2BG | 2.0 | - | - | 1.0 | - | 72.15 [d] (58.13) | 66.52 [e] (54.63) |
| MS.5Td 5G | - | 0.5 | - | 0.5 | - | 55.57 [jk] (48.18) | 49.88 [l] (44.91) |
| MSTd.5G | - | 1.0 | - | 0.5 | - | 63.12 [h] (52.59) | 59.42 [hi] (50.41) |
| MS2Td.5G | - | 2.0 | - | 0.5 | - | 70.66 [de] (57.18) | 66.68 [de] (54.72) |
| MS.5TdG | - | 0.5 | - | 1.0 | - | 64.54 [g] (53.43) | 60.48 [h] (51.03) |
| MSTdG | - | 1.0 | - | 1.0 | - | 67.42 [f] (55.18) | 61.32 [g] (51.52) |
| MS2TdG | - | 2.0 | - | 1.0 | - | 78.15 [b] (62.11) | 73.14 [b] (58.76) |
| MS.5Kn.5G | - | - | 0.5 | 0.5 | - | 44.17 [n] (41.63) | 39.66 [n] (39.02) |
| MSKn.5G | - | - | 1.0 | 0.5 | - | 52.22 [m] (46.25) | 46.42 [m] (42.93) |
| MS2Kn.5G | - | - | 2.0 | 0.5 | - | 57.98 [i] (49.57) | 51.88 [k] (46.06) |
| MS.5KnG | - | - | 0.5 | 1.0 | - | 54.19 [kl] (47.38) | 52.14 [k] (46.21) |
| MSKnG | - | - | 1.0 | 1.0 | - | 56.92 [ij] (48.96) | 50.78 [kl] (45.43) |
| MS2KnG | - | - | 2.0 | 1.0 | - | 62.55 [h] (52.25) | 60.54 [gh] (51.07) |
| MS.5BG.5N | 0.5 | - | - | 1.0 | 0.5 | 71.44 [d] (57.68) | 68.40 [d] (55.78) |
| MS.5TdG.5N | - | 0.5 | - | 1.0 | 0.5 | 75.59 [c] (60.37) | 70.44 [c] (57.04) |
| MS.5KnG.5N | - | - | 0.5 | 1.0 | 0.5 | 62.77 [h] (52.38) | 58.41 [i] (49.82) |

**Table 4.** *Cont.*

| Culture Media ▼ | Plantlet Regeneration (%) | | | | | | |
|---|---|---|---|---|---|---|---|
| | Plant Growth Regulator mg L$^{-1}$ | | | | | Mature Embryonic Axis -Derived Cell Clumps/Embryoids | Mature Cotyledon-Derived Cell Clumps/Embryoids |
| | BAP | TDZ | Kn | GA$_3$ | NAA | | |
| MSBG.5N | 1.0 | - | - | 1.0 | 0.5 | 76.82 [bc] (61.20) | 72.80 [b] (58.54) |
| MSTdG.5N | - | 1.0 | - | 1.0 | 0.5 | 84.43 [a] (66.74) | 80.19 [a] (63.55) |
| MSKnG.5N | - | - | 1.0 | 1.0 | 0.5 | 69.23 [e] (56.29) | 63.82 [f] (53.00) |
| Mean | | | | | | 64.02 (53.30) | 59.20 (50.40) |
| CD$_{0.05}$ | | | | | | 1.809 | 1.738 |

Figures in parentheses are transformed values (arc-sine transformation). Values within a column followed by different letters significantly differ at 5% probability level.

Diverse concentrations of various auxins affected relative growth rate and embryoid formation considerably (Table 1). Relative growth in higher rate was evidenced on nutrient medium MS4D (298%) followed by MS4N (292%) for liquid cultures obtained from mature embryonic axis-derived calli. The identical media also confirmed more effectiveness for suspension cultures obtained from mature cotyledon-derived calli. Other nutrient media demonstrated lesser augmentation in mass, with MS.1D (112%) and MS.1N (112%) having significantly low numbers. Embryoids/flask were obtained in higher numbers on medium MS3D (282.42 and 278.44 for mature embryonic axis- and cotyledon-derived calli, respectively), closely followed by MS4D (275.82 and 262.49 for mature embryonic axis- and cotyledon-derived calli, respectively). The remaining culture media formed embryoids in lower frequencies jointly with MS5D (124.62 and 120.75 for mature embryonic axis- and cotyledon-derived calli, respectively) with considerably low values. In terms of explant source response, mature embryonic axis-derived cell suspension (195.66) was consistently superior to mature cotyledon-derived cell suspension (190.6) ininitiating cell clumps/embryoids. After initiating cell suspensions from two different explant cultures, it was possible to differentiate the source of cultures. Cell suspensions instigated from the mature embryonic axis formed friable yellow cells with a fast expansion rate (Figure 1A), while cell suspensions acquired from mature cotyledons (Figure 1B) revealed a sluggish growth pace with whitish color.

Influences of divergent cytokinins (alone) in diverging levels on comparative growth rate and embryoid configuration are described in Table 2. The highest relative growth rate was evidenced on nutrient medium MS2Kn (178% and 169% for mature embryonic axis- and cotyledon-derived calli, respectively). Embryoids were also formed in higher frequencies with similar medium (215.38 and 211.43 for mature embryonic axis- and cotyledon-derived calli, respectively). The lower levels of cytokinins BA, Kn and TDZ (0.1–0.3 mg L$^{-1}$) resulted in a weaker stimulatory outcome on the growth of morphogenic tissues. Conversely, increased amounts of cytokinins resulted in an increment of relative growth up to 2.0 mg L$^{-1}$. With a further increment in concentration, the proportions of non-morphogenic tissues were augmented consequently. On the other hand, cytokinin kinetin promoted embryo proliferation in higher frequencies.

The combined effects of auxin in combination with cytokinin are presented in Table 3. The maximum relative growth rate was accomplished with nutrient medium MS2D.5B (386% and 372% for mature embryonic axis- and cotyledon-derived calli, respectively), followed by culture medium MS3D.5B (334% and 326% for mature embryonic axis- and cotyledon-derived calli, respectively). Embryoids/cellclumps were also evident in higher numbers with analogous nutrient medium (321.44 and 318.66 for mature embryonic axis- and cotyledon-derived calli, respectively)

*3.4. Plantlet Regeneration*

Somatic embryoids/cell clumps were subsequently transferred to regeneration medium after 6 to 8 weeks (Table 4). Shootlets in higher frequencies were recovered on MSTdG.5N

medium (84.43% and 80.19% for mature embryonic axis- and cotyledon-derived calli, respectively), followed by MS2TdG medium (78.15% and 73.14% for mature embryonic axis- and cotyledon-derived calli, respectively). The lowest presentation was displayed by regeneration medium MS.5Kn.5G (44.17% for mature embryonic axis-derived calli and 39.66% for cotyledon-derived calli).

### 3.5. In Vitro Rooting

At 3–4months of age, regenerated plantlets with 3–4 twists of leaves and 2–10 cm length were transferred on diverse formulations of MS medium for root induction. IBA in modest to a higher level supported vigorous growth of plantlets for an extended period, but root induction occurred in significantly lower frequencies, while NAA was not in any way appropriate, and the shoots could not be sustained for a longer time.

### 3.6. Acclimatization of Plantlets

Regenerants were prolifically acclimatized with a usual 75–80% survival rate under greenhouse circumstances. An arrangement of 28 °C and 65% RH displayed higher survival (~80%), followed by a combination of 30 °C and 60% RH (~75% survival rate). Approximately 65–70% of plants survived after field transplantation (data not presented). The regenerated plants, were phenotypically ordinary and true to the type.

## 4. Discussion

During the beginning of the experimentation, it was viewed that in suspension culture, calli could not crumble to provide suspension of cells, or tiny cell clumps wereprobably due to higher lignifications. For raising cell suspension cultures, transferring ~1 g friable morphogenic calli was advantageous for prolonged continuance and generation of the morphogenic cell suspension cultures. High performance from low callus inoculation frequency in the current study is in harmony with the reports for onion [38], ashwagandha [39], muskmelon [40], sarpgandha [41], chitrak [42], grape [43], soybean [31] and Indian mustard [44]. Cell clumps of 2–10 mm from 6–8-week-old cell suspension cultures were placed in fresh liquid medium. The inoculum speed was homogeneous, and several suspension cultures may be commenced by employing the tissue acquired simply from a single preliminary culture. In elevated inoculation rate, medium in a flask replenished incompletely and yielded non-embryogenic cell suspensions. During incubation, the biomass in suspension was enhanced because of cell division and expansion. This growth continued for a shorter phase as it stopped owing to the replenishment of a few factors or gathering of definite toxic metabolites in the nutrient medium. Cell growth was revived after the subculture by transferring a small aliquot of the cell suspension to a fresh medium.

In cell suspension culture, even though at lower concentrations, each of the auxins was found to instigate embryoids; such embryoids were unsuccessful in forming normal plantlets. At higher concentrations (>4.0 mg L$^{-1}$) of 2,4-D and NAA, expansion of the morphogenetic cells in suspension was impeded. Many subcultures on media supplemented with higher levels of 2,4-D caused a decrease in cell growth due to plasmolysis of cells, principally owing to the steady enlargement in the cells under the highest concentration of 2,4-D [31,38–42,44–46]. In contrast, at lower magnitudes of NAA, a lower-bustle auxin generated fast-developing condensed calli and initiated cell suspensions in lower ratios. Analogous verdicts have also been reported by Tiwari et al. [38] for onion, Jhankare et al. [39,47] for ashwagandha, Bariwa et al. [40] for muskmelon, Patidar et al. [42] for chitrak, Sharma et al. [41] for grape, Mishra et al. [31] for soybean and Shyam et al. [44] for mustard cell suspension cultures.

Conversely, for the initiation of cell suspension from the mature embryonic axis, the reaction of auxins was reversed as NAA supported the culture of single cells and 2,4-D demonstrated early toxicity to cultures. Later on, initiating two types of cell suspensions from mature cotyledons and embryonic axis was likely to differentiate the origin of cultures. Cell suspensions initiated from the mature embryonic axis directly formed chlorophyllous

cells with a rapid growth rate, whereas cell suspensions acquired from mature cotyledons revealed a sluggish growth rate with white color. The initiation trends of cell cultures were deepened based on types of auxin. Growth regulator NAA showed a beneficial effect on the cell growth of suspension cultures raised from mature embryos that kept their fast-growing phenol-free status long-term. Most of the regenerants were confirmed to be phenotypically stable under field testing. This finding was in accordance with Rugkhla and Jones [48], Bele et al. [49] and Tripathi et al. [14], who stated that the auxins, especially NAA and 2,4-D, play an imperative role in the regulation of somatic embryogenesis in sandalwood.

Cytokinins BA, Kn and TDZ (0.1 mg L$^{-1}$) at the lower concentration level resulted in the minor stimulatory outcome of morphogenetic tissues' escalation. In contrast, elevated concentrations of cytokinins yielded an increment of relative growth up to 2.0 mg L$^{-1}$. At higher concentrations of cytokinins, the proportion of non-embryogenic tissues increased consequently. Alternatively, cytokinin kinetin promoted embryo proliferation in higher frequencies. However, nutrient media amended only with cytokinins have been proven to be infective for the growth and development of suspension cultures compared to auxins, as speedy growth and development of morphogenic cultures could be attained devoid of their existence. This incident is in agreement with the pronouncements of Tiwari et al. [38] for onion, Jhankare et al. [39] for ashwagandha, Bairwa et al. [40] formuskmelon, Sharma et al. [43] for grape and Shyam et al. [44] for mustard since they reported that it is not essential or advantageous to include cytokinins for initiating morphogenic cell suspension cultures. However, in many species, cytokinins were found effective for somatic embryo induction [50,51].

Even strengthening of nutrient media with auxins and cytokinins disjointedly did not sustain the commencement of cell suspension cultures. Hence, diverse combinations of auxin in combination with a cytokinin were also tested. The liquid medium supplemented with 2.0–3.0 mg L$^{-1}$ 2,4-D and a lower level of BAP (0.5 mg L$^{-1}$) was the most effective for establishing cell suspension cultures. In the early 2–3 subcultures, this culture medium produced loose, friable clumps masses from the inoculated morphogenic calli, which outline delicate light cell suspensions of yellow-cream color. In a successive subculturing, the 2,4-D level was trimmed down to 1.0–2.0 mg L$^{-1}$, which sustained the speedy advancement of embryoids. Analogous findings were documented by Tiwari et al. [38] for onion, Jhankare et al. [39] for ashwagandha, Bairwa et al. [40] for muskmelon, Sharma et al. [41] for grape and Mishra et al. [31] for soybean suspension cultures. Although medium amended with a higher proportion of kinetin in permutation with NAA proved more effective for enhancing biomass of suspension cultures, it failed to convert into higher numbers of cell clumps/embryoids. Nutrient medium supplemented with NAA and BA was unable to initiate cell suspension cultures despite creating bulky compact calli of green color. Cells failed to detach in the suspension culture to shape refined suspension, suggesting that cell suspension culture instigation is auxin-dependent.

On the other hand, a much lower cell division rate was observed without plant growth regulators. Auxin dependence conforms to the verdicts of Bairwa et al. [40] for muskmelon, Patidar et al. [42] for chitrak, Sharma et al. [43] for grape and Shyam et al. [44] for mustard cell suspension cultures. However, optimal concentration and combination of 2,4-D and Kn and/or BAP depended upon the nature of the explant sources from which calli were derived for cell suspension culture establishment.

Embryoids/cell clumps obtained from embryogenic cell suspension culture were subsequently transferred to a regeneration medium containing different plant growth regulator regimes. PGRs 2,4-D and NAA alone did not sustain complete plant regeneration in suspension culture during this study. A combination of 1.0 mg L$^{-1}$ cytokinin TDZ with 1.0 mg L$^{-1}$ GA$_3$ and 0.5 mg L$^{-1}$ NAA supported higher plantlet regeneration. It enhanced shoot proliferating ability more than medium with a cytokinin still at the higher level, and an auxin reacted inadequately. Hence, it was proven that shootlet regeneration is decided by quantitative communication, i.e., ratios compared to the total concentration of essences contributing to growth and development. This is in harmony with the results

of Tiwari et al. [38] for onion, Jhankare et al. [39] for ashwagandha, Bairwa et al. [40] for muskmelon and Mishra et al. [31] for soybean cell suspension cultures. Similar results were also obtained by Tiwari and Tripathi [52] for soybean explant cultures, Sharma et al. [53] for *Glycyrrhiza glabra* (L.), Patidar et al. [54] for *Emblica officinalis* (Gaertn.), Vibhute et al. [55] for *Citrus* spp., Bele et al. [23,49], Tripathi et al. [14] for sandalwood and Tripathi et al. [56] for *Gladiolus hybridus* (L.) explant cultures.

Furthermore, the protocol explained in this study necessitated diverse plant growth regulator concentrations and combinations from those formerly documented to acquire the most normal somatic embryoids followed by their complete germination. Rao and Bapat [57] employed media supplemented with IAA and BAP for multiplication and conversion to shootlets and observed the dilemma of low conversion incidence. During the present experimentation, if mature somatic embryoids/cell clumps were not germinated in media amended with $GA_3$, several abnormal somatic embryoids emerged, causing a low conversion into shootlets.

Different hormonal permutations and combinations were attempted as an add-on to MS semisolid and liquid media. Shockingly, the expurgated shootlets did not form roots in any of the amended rooting media, even though there was an extended period after adopting diverse culture regimes. Plantlets were entrenched on MS semisolid rooting medium with diverse levels of NAA, IBA and Kn alone plus IBA in association with BA, Kn NAA and $GA_3$. Combination of IBA with kinetin induced initiation of friable white calli at the slash end; nevertheless, the calli did not shape into the root. IBA applied in combination with BA somewhat persuaded adventitious shootlet initiation in place of roots. Semi-solid medium with diverse levels of IBA in combination with BA/Kn and $GA_3$ also did not initiate root formation in the shoots. These reports have conformity with Sarangi et al. [19], Bele et al. [23,49] and Tripathi et al. [14], who failed to induce in vitro rooting in sandalwood despite a long and cyclical endeavor.

In terms of the explant source for raising callus culture to establish cell suspension cultures, considerable variability was observed between the two explants sources. Mature embryonic axis-derived calli were found consistently superior to mature cotyledon-derived calli for most of the phases of culture investigated. Performances of different explant sources in terms of suspension culture initiation and establishment have been addressed by Tiwari et al. [38] for onion, Jhankare et al. [39] for ashwagandha, Bairwa et al. [40] for muskmelon and Mishra et al. [31] for soybean cell suspension cultures.

The protocol established from present experimentations necessitated diverse hormonal amalgamations and concentrations from those formerly described by researchers. In the present experiment, TDZ and/or BA combined with 2,4-D could impulsively initiate embryogenesis at remarkable magnitudes and with better reproducibility. Present results are comparable to the verdict of Rugkhla and Jones [48], as they also acquired 100% somatic embryo initiation with supplementation of TDZ alone and in association with 2,4-D. Mature somatic embryos required the addition of $GA_3$ for germination, conversion and elongation of regenerants; otherwise, many deformed somatic embryos were acquired, resulting in a low conversion into shootlets. In summary, the current findings authenticate the reality that these techniques may be used to synthesize the secondary metabolites for use in the pharmaceutical business. The multiplication of friable morphogenic tissue or cyclic embryogenesis is useful as basic material for genetic transformation and selection of valuable mutants at the cell level.

**Author Contributions:** Data curation, M.K.T.; formal analysis, M.K.T., S.T. (Sushma Tiwari), D.B. and S.S.; investigation, M.K.T., G.T., D.B., N.M. and R.P.P.; methodology, M.K.T.; projectadministration, M.K.T.; resources, M.K.T.; software, M.K.T. and S.S.; writing—original draft, M.K.T. and N.T.; writing—review and editing, M.K.T., N.T., S.T. (Sharad Tiwari) and S.S. All authors have read and agreed to the published version of the manuscript.

**Funding:** This research received no external funding.

**Institutional Review Board Statement:** Not applicable.

**Informed Consent Statement:** Not applicable.

**Data Availability Statement:** The data supporting the findings of this study are available from thecorresponding author upon reasonable request.

**Conflicts of Interest:** The authors declare no competing interest.

## Abbreviations

BA-6    benzylaminopurine;
2,4-D    2,4-dichlorophenoxyacetic acid;
GA$_3$    gibberellic acid;
IBA    indole-3-butyric acid;
Kn    kinetin;
MS    Murashige and Skoog medium;
NAA    α-naphthalene acetic acid;
PGR    plant growth regulator;
TDZ    thidizuron.

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
