# Peer review of "Optimization of Different Factors for Initiation of Somatic Embryogenesis in Suspension Cultures in Sandalwood (Santalum album L.)"

_horticulturae, doi:10.3390/horticulturae7050118_

Round 1

Reviewer 1 Report

1). Manuscript ID: Horticulturae-1206286

2). Manuscript Title: Optimization of different factors for initiation of somatic embryogenesis in suspension cultures in sandalwood (Santalum album L.)

3). Comments: Please revise the manuscript based on the following comments.

General comments

--Extensive English changes required. Please edit the manuscript thoroughly before submission.

--Please follow the Journal format while revising the manuscript.

--Add scientific authority at the end of binomial names of all species when they are mentioned for the first time in the manuscript.

--Include full forms of all abbreviations/acronyms mentioned in the manuscript.

Specific comments

--Figure1A to P: Please include scale bar 

--Line 79: Include the cultivar used for explants

--Line 92: Correction: “2,4-D”

--Line 185: Correction: “Figs. 1A-B”

--Line 189: What are high-flying nuclei?

--Line 207: Correction: “greenhouse”, “net house”

--Lines 324: Correction: “ineffective” for growth and development

Reviewer 2 Report

Some additions need to be made to the article:
abstract
line 30: GA3 concentration not specified
materials
it is not indicated how hormones were introduced into a solid medium for regeneration
picture 1: no scale
A, B, G, F, J, L photos are low resolution and not informative. Better to replace with other photographs.
There are misprints in the text, you need to check the text.

Round 2

Reviewer 1 Report

1). Manuscript ID: Horticulturae-1206286

2). Manuscript Title: Optimization of different factors for initiation of somatic embryogenesis in suspension cultures in sandalwood (Santalum album L.)

3). I'm satisfied with the revised manuscript. Thank you!

Reviewer 2 Report

More information on in vitro production of somatic embryoids for sandalwood should be added in the introduction. Plant growth and development regulators have been used for cultural media. It is necessary to clarify when (before or after autoclaving) and in what form (dry or solution) the regulators were added.
